# 3DPCP-Net: A Lightweight Progressive 3D Correspondence Pruning Network for Accurate and Efficient Point Cloud Registration

## ABSTRACT

Accurately identifying correct correspondence (inlier) within initial ones is pivotal for robust feature-based point cloud registration. Current methods typically rely on one-shot 3D correspondence classification with a single coherence constraint to obtain inlier. These approaches are either insufficiently accurate or inefficient, often requiring more network parameters. To address this issue, we propose a lightweight network, 3DPCP-Net, for fast and robust registration. Its core design lies in progressive correspondence pruning through mining deep spatial geometric coherence, which can effectively learn pairwise 3D spatial distance and angular features to progressively remove outlier (mismatched correspondence) for accurate pose estimation. Moreover, we also propose an efficient feature-based hypothesis proposer that leverages the geometric consistency features to generate reliable model hypotheses for each reliable correspondence explicitly. Extensive experiments on 3DMatch, 3DLoMatch, KITTI and Augmented ICL-NUIM demonstrate the accurate and efficient of our method for outlier removal and pose estimation tasks. Furthermore, our method is highly versatile and can be easily integrated into both learning-based and geometry-based frameworks, enabling them to achieve state-of-the-art results. The code is provided in the supplementary materials.

## CCS CONCEPTS

• **Theory of computation** → **Computational geometry**; • **Computing methodologies** → **Matching**.

## KEYWORDS

Point Cloud Registration, Progressive Correspondence Pruning, Deep Spatial Geometric Coherence, Hypothesis Proposer, Lightweight, Accurate and Efficient

## 1 INTRODUCTION

Point cloud registration serves as a fundamental component in numerous 3D computer vision applications, including 3D reconstruction [7], simultaneous localization and mapping (SLAM) [12], autonomous driving [27]. Its aims to align two 3D scan fragments captured from different views. The canonical solution commences

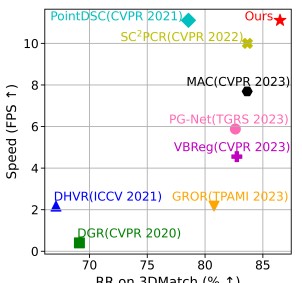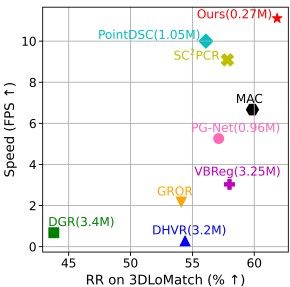

**Figure 1: Experimental results on 3DMatch and lower-overlap 3DLoMatch. Our method outperforms other methods in terms of registration accuracy while maintaining fast speed and lightweight.**

by establishing feature correspondences leveraging 3D local features [32, 39], subsequently estimating the optimal rigid transformation encompassing both 3D rotation and translation. However, due to the non-uniform data quality (*e.g.*, noise distribution, repetitive structures, domain gaps between different sensors), limited overlap and limitations of existing 3D descriptors, pose estimation suffers from numerous outliers in the correspondences leading to inaccurate or incorrect 3D registration.

3D registration focusing on outlier removal can be divided into geometry-based and deep learning-based methods. For geometry-based methods [5, 14, 52], RANSAC [14] leverages an iterative sampling strategy for outlier removal. Its effectiveness deteriorates with increasing outlier burden, necessitating more iterations to achieve convergence and leading to higher computational cost. Recently, $SC^2$-PCR [5] and MAC[52] follow the hypothesis verification concept from RANSAC to obtain acceptable results. $SC^2$-PCR presents a second order spatial compatibility measure to enhance the robustness against outliers. MAC presents a maximal clique constraint to mine more local information. They utilize a single consistency measure (spatial length) to compute the affinity between initial correspondences and lack contextual information, resulting in decreased accuracy and efficiency in lower-overlap scenes as shown in Fig. 1. Deep learning-based methods [1, 8, 20, 24, 26, 29, 40] typically define outlier removal as the inlier/outlier classification problem. They utilize the compatibility measures for putative correspondence to output the probability of inliers, thereby removing low probability outliers. The compatibility measure mostly miss the essential 3D spatial coherence. Among them, PointDSC [1], a relatively reasonable method, relies on a non-local network driven by spatial length consistency to capture long-range context. However, as shown in

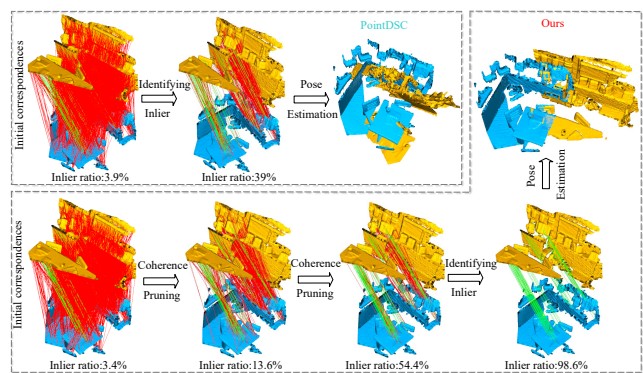

**Figure 2: Comparison of one-shot 3D matching classification and progressive 3D correspondence pruning by coherence mining for pose estimation on 3DLoMatch. The green and red line represent inliers and outliers, respectively.**

Fig. 2, the one-shot correspondence classification manner with single coherence still yields limited pose estimation, especially for scenes with high initial outlier ratios (lower-overlap often > 95%). Furthermore, as illustrated in Fig. 1, these deep learning-based methods require a considerable amount of network parameters and struggle to balance the accuracy and efficiency of registration.

In this paper, to alleviate the above problems, we first propose to progressively prune the set of correspondences, rather than a one-shot classification of initial correspondences. Since most outliers are expected to be filtered out after the progressive pruning, this strategy allows us to identify reliable inliers among the remaining entities (as shown in Fig. 2), which leads to accurate pose estimation. To ensure an accurate pruning strategy, it is essential to differentiate between inliers and outliers as precisely as possible. Instead of individually distinguishing each correspondence, we propose Deep Geometric Consistency (DGC) block to integrate long-range contextual and pairwise 3D spatial geometric consistency constraints. Benefiting from the invariance of spatial distance and angle under rigid transformation between inliers and the feature similarity, DGC can obtain more representative features for each correspondence, facilitating accurate inlier/outlier differentiation. Secondly, we propose a feature-based hypothesis proposer that leverages the high-dimensional spatial geometric consistency features extracted above and spatial geometric consistency searching to efficiently generate multiple reliable hypotheses. The optimal hypothesis with the most support from the correspondences is selected as the final alignment. Explicitly and progressively pruning the initial correspondences, as opposed to purely using deeper networks, does not result in an increase in network parameters. Meanwhile, explicitly generating hypotheses instead of implicitly computing the transformation using regression variables [39] contributes to the fast and accurate estimation of the optimal transformation. Overall, our have four-fold of contributions:

- We propose a lightweight progressive 3D correspondence pruning network to mitigate the influence of massive outliers for accurate and efficient point cloud registration.

- We introduce a DGC learning block to examine the compatibility of two correspondences by exploring deep feature similarity and pairwise spatial distances and angles, thereby facilitating the correspondence pruning process.
- Compared with state-of-the-art methods, our method achieves favorable performance on 3DMatch, 3DLoMatch, KITTI and Augmented ICL-NUIM datasets.
- Our approach can be easily integrated into other deep learning-based or geometry-based frameworks, such as PointDSC and SC$^2$-PCR. Through experiments, we demonstrate that combining them with our method produces superior results.

## 2 RELATED WORK

### 2.1 3D Feature Matching

Classic Iterative Closest Point [3] and its variants [4, 47], which rely on a good initial pose, generate correspondences based on Euclidean distance in coordinate space. In contrast, some methods establish correspondences by matching local descriptors with recognizable information in feature space. Geometry local descriptor methods involve providing numerical results [10, 13] or representing point cloud information as histograms [34, 54]. Learning 3D local descriptors has been widely studied. PointNet [30, 31] is the first network to directly extract features on input point clouds. It solves the problems of disorder, replacement invariance, and rotational invariance for point clouds using a multilayer perceptron, symmetric function (maximal pooling), and transformation network. Recent learning descriptors can be divided into local patch-based [11, 17, 50] and convolution-based [9, 32, 42, 51] according to the characteristics of their feature extraction. These methods have achieved significant performance improvements, but they struggle to establish a completely outlier-free correspondence set. They rely on outlier removal method to estimate robust rigid transformation.

### 2.2 Outlier Removal

**Geometry-based methods.** Geometry methods can be categorized as score-based and label-based. Score-based methods rank 3D correspondences based on consistency score and the top-K correspondences are selected as the inliers. GTM [33] seeks global consistency constraints among surface points during local operations. MV [49] introduces a mutual voting approach for sorting correspondences, achieving reliable correspondences scoring results. LT-GV [46] develops a loose-tight geometric voting method that uses both loose and tight geometric constraints in the graph to score correspondences. CV [48] employs a consistency voting method that utilizes distance and spatial constraints to rank 3D correspondences; it scores each correspondence by examining the consistency with a predefined voting set. The classic RANSAC [14] and its variants [23, 38] are popular outlier label-based methods that iteratively randomly sample the smallest subset in the sampling step. They have slow convergence and low accuracy with relatively high outliers. Graph-cut RANSAC [2] introduces graph-cut technique for better local optimization. SM [25] establishes a compatibility graph through length consistency and obtains an inlier set by identifying the main cluster of the graph. FGR [55] and TEASER [44] can reject outliers using the Geman-McClure cost function. Recently, SC$^2$-PCR [5] introduced a second-order spatial

compatibility measure to quantify the affinity between correspondences. MAC [52] introduced a maximal clique constraint that can extract more local information in the compatibility graph. These geometric methods often ignore the spatial angle coherence and contextual information.

**Deep learning-based methods.** Similar to the 2D feature matches [53], they attempt to design a unique deep learning classifier. 3DReg-Net [29] and DGR [8] train an end-to-end neural network using operators such as sparse convolution and point-by-point MLP [30] to classify assumed correspondences. PointDSC [1], serving as our baseline, proposes a non-local inlier classifier guided by length consistency, followed by robust alignment via neural spectral matching. DHVR [24] utilizes deep Hough voting to establish consensus between correspondences from the Hough space, thus predicting the final transformation. DetarNet [6] proposes a decoupling solution for translation and rotation based on 3DRegNet. PG-Net [40] employs a grouped dense fusion attention feature embedding module to enhance the representation of inliers and significant channel–spatial information. VBReg [20] proposes a novel framework based on variational non-local networks to capture long-range dependencies in geometric context for inlier/outlier distinction. The learning methods often overlook distance and angle consistency constraints in the 3D domain. However, they all utilize one-shot correspondence classification and ignore the constraints of spatial geometric consistency (pairwise angles and distances). In contrast, our network progressively prunes the initial correspondences. This simple strategy can effectively solve the problem of excessive outliers in low-overlap scenes without increasing the network parameters. In addition, our method explicitly integrates the geometric consistency between inliers into the contextual information to accurately and progressively removal outliers.

## 3 METHOD

### 3.1 Problem Formulation

For the partially overlapping source point cloud $\mathbf{P}^s = \{\mathbf{p}_i^s \in \mathbb{R}^{N_s \times 3}\}$ and target point cloud $\mathbf{P}^t = \{\mathbf{p}_i^t \in \mathbb{R}^{N_t \times 3}\}$ to be aligned, we first extract local features for them using geometric or learned 3D descriptors. Then, for each point in $\mathbf{P}^s$, we find its nearest-neighbor local feature in $\mathbf{P}^t$ to generate an initial correspondence set $\mathbf{C} = \{\mathbf{c}_1, \mathbf{c}_2, ..., \mathbf{c}_N\} \in \mathbb{R}^{N \times 6}$, where $\mathbf{c}_i = (\mathbf{p}_i^s, \mathbf{p}_i^t)$. Our aim is to identify an inlier/outlier label for all $\mathbf{c}_i$ (*i.e.*, $w_i = 0$ or 1) and estimate the optimal rigid transformation rotation matrix $\mathbf{R} \in \mathbb{R}^{3 \times 3}$ and translation vector $\mathbf{t} \in \mathbb{R}^3$ between $\mathbf{P}^s$ and $\mathbf{P}^t$ using $\boldsymbol{w} = \{w_1, w_2, ..., w_N\}$ weighted least squares fitting [3],

$$\mathbf{R}, \mathbf{t} = \underset{\mathbf{R}, \mathbf{t}}{\arg\min} \sum_i^{|\mathbf{C}|} w_i \parallel \mathbf{R}\mathbf{p}_i^s + \mathbf{t} - \mathbf{p}_i^t \parallel^2 . \tag{1}$$

Eq. 1 can be solved using SVD [37]. In other words, accurately predicting the $\boldsymbol{w}$ is crucial for aligning $\mathbf{P}^s$ and $\mathbf{P}^t$. However, the presence of excessive outliers in $\mathbf{C}$ makes it difficult to accurately predict $\boldsymbol{w}$ for all $\mathbf{c}_i$ in a one-shot manner. In this paper, we suggest progressively prune $\mathbf{C}$ into a candidate subset $\mathbf{C}_m \in \mathbb{R}^{N_m \times 6}$ ($N_m$ representing the number of correspondences at this time) to mitigate the impact of outliers. More accurate pose estimates $\mathbf{R}'$ and $\mathbf{t}'$ can be obtained by the pruned subset $\mathbf{C}_m$.

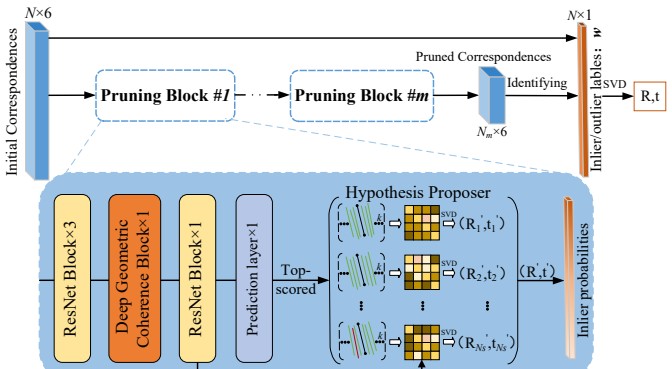

**Figure 3: The framework of our method. By employing an progressive pruning strategy, 3DPCP-Net takes $N \times 6$-dimensional initial correspondences as input and outputs the optimal rotation transformation $(\mathbf{R}, \mathbf{t})$ and $N \times 1$-dimensional inlier/outlier lables. Deep Geometric Consistency (DGC) block and feature-based Hypothesis Proposer are two key components of each pruning block, which respectively perform compatibility feature learning and model fitting.**

### 3.2 3DPCP-Net Framework

Our proposed progressive 3D correspondence pruning network (3DPCP-Net) is illustrated Fig. 3. The crucial innovation of our framework is the progressive pruning of the initial correspondences via inlier probabilities to obtain a more reliable subset of correspondences. In this process, we achieve progressive pruning by sequential "pruning" blocks to filter out some outliers. The core of 3DPCP-Net is $m$ coherence pruning blocks. Each pruning block is composed of our proposed DGC block, Hypothesis Proposer and existing structure ResNet blocks [18]. The pruning block is primarily utilized to compute the inlier probability for progressive pruning. Specifically, each pruning block first extracts an intermediate feature representation $\mathbf{f_i}$ via a series of ResNet blocks [18] for each correspondence. Then, the proposed DGC block is leveraged to obtain the high-dimensional geometric consistency features of each $\mathbf{c}_i$. Subsequently, a ResNet block and MLP layer are employed to estimate the confidence score $e_i$ for each $\mathbf{c}_i$, and the top-$N_S$ most reliable correspondences are selected. Subsequently, our proposed Hypothesis Proposer is utilized to obtain the optimal transformation $\mathbf{R}'$ and $\mathbf{t}'$. The final inlier probabilities of each $\mathbf{c}_i$ is represented by the error $\parallel \mathbf{R}'\mathbf{p}_i^s + \mathbf{t}' - \mathbf{p}_i^t \parallel$. Ultimately, after passing $m$ pruning blocks, the inlier/outlier labels $w_i = [\parallel \mathbf{R}'\mathbf{p}_i^s + \mathbf{t}' - \mathbf{p}_i^t \parallel < \varepsilon]$ are identified from the initial correspondences, and the final $\mathbf{R}$ and $\mathbf{t}$ are estimated with reference to Eq. 1. Where $\varepsilon$ denotes an inlie threshold and $[\cdot]$ represents the Iverson bracket. Below, we will provide a detailed introduction to the DGC block and Hypothesis Proposer within the pruning blocks.

### 3.3 Deep Geometric Coherence

To distinguish the inliers and outliers, it is crucial to mine the geometric consistency information [45] of inliers in the 3D domain. As shown in Fig. 5(a), an important observation is that the inliers (*e.g.*,

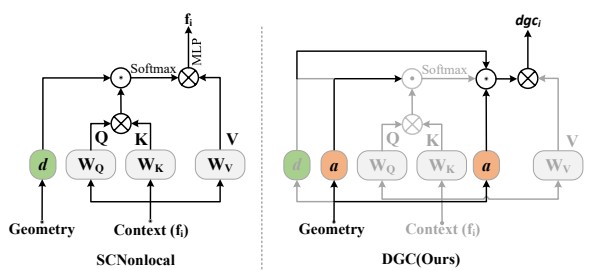

Figure 4: Illustration of different computation in the SCNonlocal [1] and DGC.

$c_2$ and $c_3$) are geometrically compatible, *i.e.* the spatial distance $d_{23} = d'_{23}$ and spatial angle $\alpha_{23} = \alpha'_{23}$. And outliers are incompatible with either inliers or outliers (*e.g.*, $c_1$ and $c_2$, $c_4$ and $c_5$ ), *i.e.* $\alpha_{12} \neq \alpha'_{12}$ and $d_{45} \neq d'_{45}$. Additionally, the commonly used context feature similarity can also be leveraged to distinguish inliers and outliers. Inspired by these considerations, we propose a deep geometric coherence measure to examine the compatibility between two correspondences.

We first consider the distance and angle constraints, which are rotation-invariant compatibility measures under rigid transformation. Given two correspondences $(c_i, c_j)$, the distance and angle constraints are respectively defined as:

$$d_{ij} = \big| \, \| \mathbf{p}_i^s - \mathbf{p}_j^s \| - \| \mathbf{p}_i^t - \mathbf{p}_j^t \| \, \big|, \tag{2}$$

$$\alpha_{ij} = \big| \mathrm{acos}(\mathbf{n}_i^s \cdot \mathbf{n}_j^s)\frac{180}{\pi} - \mathrm{acos}(\mathbf{n}_i^t \cdot \mathbf{n}_j^t)\frac{180}{\pi} \big|, \tag{3}$$

where $\mathbf{n}$ denotes the local normal (*e.g.*, tangent space) for each point.

By integrating these two constraint conditions, we define the geometric coherence measure of $(c_i, c_j)$ as follows:

$$gc_{ij} = \exp(-\frac{d_{ij}^2}{\sigma_d^2} - \frac{\alpha_{ij}^2}{\sigma_\alpha^2}), \tag{4}$$

where a clearly ranges within (0,1], coinciding with 1 only when both distance and angle constraints are fully satisfied. $\sigma_d$ and $\sigma_\alpha$ are parameters controlling the sensitivity to differences in distance and angle, respectively. $(c_i, c_j)$ is considered incompatible if the difference in distance $d_{ij}$ exceeds $\sigma_d$ or the difference in angle $\alpha_{ij}$ exceeds $\sigma_\alpha$, resulting in a smaller value of $gc_{ij}$. When $(c_i, c_j)$ is geometrically compatible in spatial, $gc_{ij}$ can take on a larger value. Observing Fig. 5, there is ambiguity when inlier $c_2$/outlier $c_1$ and inlier $c_3$/outlier $c_4$ satisfy just length consistency. The angle constraint provides a possibility to alleviate the ambiguity issue. Therefore, these two constraints are complementary to each other.

DGC leverages self-attention to integrate long-range contextual information and geometric consistency constraints, which is a common practice [1, 32]. Specifically, as illustrated in Fig. 4 right, we update the geometric coherence measure for each correspondence using the following equation:

$$dgc_i = \sum_j^{|\mathbf{C}|} \mathrm{softmax}_j(\mathbf{Q} \otimes \mathbf{K} \odot gc) \odot gc \otimes \mathbf{V}, \tag{5}$$

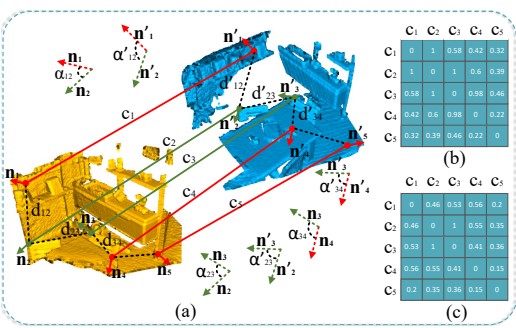

Figure 5: (a): Inlier pairs satisfy the geometric coherence. d and $\alpha$ denote spatial distances and angles. n is the local normal. (b) and (c): The length (distances) and geometric (distances and angles) consistency compatibility matrix of (a), respectively.

where $\mathbf{Q}$, $\mathbf{K}$, $\mathbf{V}$ are a linear projection for intermediate feature $\mathbf{f}_i$. $\mathbf{Q} \otimes \mathbf{K}$ is the feature similarity term. $gc$ is the weight matrix of $gc_{ij}$.

Note that PointDSC propose SCNonlocal [1], as shown in Fig. 4 left, to integrate spatial length consistency with a non-local network for feature encoding. However, the angle consistency clues inherent in the original geometry are entirely ignored. To this end, DGC generates geometric constraints $gc_{ij}$ that incorporate angle consistency. At the local level, $gc_{ij}$ is combined with feature similarity for feature augmentation. On the global level, it is employed to learn a geometric rotation-invariant representation of long-range cross-frame context aggregation, which facilitates the precision discrimination between inliers and outliers. The learned features $\boldsymbol{dgc}_i$ of DGC for each correspondence will be utilized for confidence score $e_i$ computation and the subsequent Hypothesis Proposer.

### 3.4 Hypothesis Proposer

We follow the hypothesize-and-verify pipeline of RANSAC. For $N_S$ reliable correspondences, we perform the top-$k$ second order spatial compatibility [5] searching to obtain corresponding consensus sets $\mathbf{C}'_j \subseteq \mathbf{C}_m$ ($| \, \mathbf{C}'_j |= k$, $j = 1, 2, ..., N_S$) in the geometric feature space learned from the DGC block. Then, the least squares fitting is performed on each consensus set to obtain rotation and translation transformation sets $\{(\mathbf{R}'_1, \mathbf{t}'_1), (\mathbf{R}'_2, \mathbf{t}'_2), ..., (\mathbf{R}'_{N_S}, \mathbf{t}'_{N_S})\}$. Finally, the optimal $(\mathbf{R}', \mathbf{t}')$ is selected among $N_S$ hypothesis transformations.

In specific, we construct a compatibility matrix $\mathbf{T}$ for each $\mathbf{C}'_j$. Each $T_{ij}$ measures the compatibility between correspondences $\mathbf{c}'_i$ and $\mathbf{c}'_j$ defined as:

$$T_{ij} = [1 - \frac{1}{\sigma_f^2}\|\bar{dgc}_i - \bar{dgc}_j\|^2], \tag{6}$$

where $\bar{dgc}_i$ and $\bar{dgc}_j$ are the L2-normalized feature vectors of $\boldsymbol{dgc}_i$ and $\boldsymbol{dgc}_j$, $\sigma_f$ is a parameter that controls the sensitivity to feature differences form DGC.

Following [25], the dominant cluster of matrix $\mathbf{T}$ is formed by the statistics of inliers, thus naturally interpreting the leading eigenvector of $\mathbf{T}$ as the inlier probability. The leading eigenvector $v \in \mathbb{R}^k$ is efficiently computed using the power iteration algorithm [28].

We utilize $v$ as weights for the least-squares fitting to estimate the transformation $(\mathbf{R}'_j, \mathbf{t}'_j)$ for each consensus set $\mathbf{C}'_j$ with reference to Eq. 1.

The last stage of the pruning block selects the best hypothesis $(\mathbf{R}', \mathbf{t}')$ among the $N_S$ rotation transformations $(\mathbf{R}'_j, \mathbf{t}'_j)$. The criterion for selecting the best transformation is based on the number of initial correspondences $\mathbf{C}_m$ satisfied by each transformation,

$$\mathbf{R}', \mathbf{t}' = \underset{\mathbf{R}'_j, \mathbf{t}'_j}{\operatorname{argmax}} \sum_{i}^{|\mathbf{C}_m|} [\| \mathbf{R}'_j \mathbf{p}_i^s + \mathbf{t}'_j - \mathbf{p}_i^t \| < \varepsilon], \qquad (7)$$

where $\varepsilon$ denotes an inlie threshold and $[\cdot]$ represents the Iverson bracket.

## 3.5 Loss Function

A hybrid loss function is used to optimize our proposed approach:

$$L = \mu L_{class} + L_{match}, \qquad (8)$$

where $L_{class}$ represents the binary cross-entropy loss [8, 29] to individually supervise each correspondence. $L_{match}$ denotes the matching loss to supervise pairwise correspondences. $\mu$ is a hyper-parameter used to balance two losses.

$$L_{class} = \sum_{n}^{m} \text{BCE}(\mathbf{e}_n, \mathbf{w}_n^*), \qquad (9)$$

where $m$ is the number of pruning blocks. $\mathbf{e}_n, \mathbf{w}_n^* \in \mathbb{R}^{|\mathbf{C}_n|}$ and $\mathbf{C}_n$ is the confidence score, ground-truth inlier/outlier labels and initial correspondences in $n$-th pruning block, respectively. Where

$$w_i^* = [\| \mathbf{R}^* \mathbf{p}_i^s + \mathbf{t}^* - \mathbf{p}_i^t \| < \varepsilon], \qquad (10)$$

where $\mathbf{R}^*$ and $\mathbf{t}^*$ denote the ground-truth rotation matrix and translation vector, respectively.

$$L_{match} = \sum_{n}^{m} \frac{1}{|\mathbf{C}_n|^2} \sum_{ij} (T_{ijn} - T_{ijn}^*)^2, \qquad (11)$$

where $T_{ijn}$ and $T_{ijn}^*$ are $T_{ij}$ and $T_{ij}^*$ in $n$-th pruning block. $T_{ij}$ is defined in Eq. 6. $T_{ij}^*$ indicates that both $\mathbf{c}_i$ and $\mathbf{c}_j$ in $T_{ij}$ are inliers.

## 4 EXPERIMENTS

### 4.1 Experimental Setup

**Datasets.** For pairwise registration, we used the indoor dataset 3DMatch [51] and 3DLoMatch [19] and the outdoor dataset KITTI [15]. Following [8] , for 3DMatch, we selected 2186 pairs of partially overlapping point cloud fragments for training, with 1623 pairs used for training from 8 scenes. The performance of the method is further evaluated using the 3DLoMatch benchmark that contains 1781 pairs of low overlapping point clouds. 3DLoMatch is a subset of 3DMatch, where the overlap ratio ranges from 10% to 30%, which is highly challenging. For KITTI, we selected 1358 pairs for training and 555 pairs for testing from 11 scenes. For multi-way registration, we utilized the Augmented ICL-NUIM dataset [7], which includes four indoor environmental scenes: two living room sequences and two office sequences.

**Evaluation Criteria.** For pairwise point clouds, we calculate the rotation error (RE) and translation error (TE) separately. We report the registration recall (RR) at an error threshold to evaluate

the registration results on the dataset. Following [8], successful registration is considered when RE $\leq 15°$ and TE $\leq 30$ cm for the 3DMatch and 3DLoMatch dataset, and RE $\leq 5°$ and TE $\leq 60$ cm for the KITTI dataset. Following [1], we also reported the inlier precision (IP, %), inlier recall (IR, %), and F1-measure (F1, %) to evaluate the outlier removal results. For multi-way registration, following [8], we reported the absolute trajectory error (ATE, cm).

**Implementation Details.** For 3DMatch/3DLoMatch and KITTI datasets, we construct voxel grids with resolutions of 5cm and 30cm, respectively, to downsample the point clouds. We randomly extract 1000 initial correspondences using FPFH [34] and FCGF [9] descriptors for training. We set the batch size to 16 point cloud pairs. Additionally, for Augmented ICL-NUIM, we also use a 5cm voxel grid for downsampling and extract FPFH descriptors. For each pruning the top 25% correspondences are selected as input for the next pruning block. When learning the DGC measure, we make parameter $\sigma_d$ as 0.1m and $\sigma_\alpha$ as 15° for indoor scenes, while for outdoor scenes, we set parameter $\sigma_d$ as 0.6m and $\sigma_\alpha$ as 5°. The number of reliable correspondences ($Ns$) is set to $0.2 * N$, where $N$ is the assumed number of correspondences ($N_m$) at that time. For each reliable correspondence, we select 40 nearest neighbors ($k = 40$). When estimating rigid transformations, we allow $\sigma_f$ to be learned through the network, with parameter $\varepsilon$ set to 0.1cm for indoor scenes and 0.6cm for outdoor scenes. The hyperparameter $\mu$ is set as 3. We trained the network for 50 epochs using the ADAM optimizer with an initial learning rate of 0.0001 and an exponential decay factor of 0.99. Our work is implemented in PyTorch. All experiments are conducted on an RTX3090 graphics card.

## 4.2 Results on Indoor Scenes

We first compare our method on 3DMatch dataset with 14 baselines: FGR [55], SM [25], TEASER [44], GC-RANSAC [2], RANSAC [14] (1k, 10k and 100k iterations, respectively)), GROR [43], SC$^2$-PCR [5], MAC [52], 3DRegNet [29], DGR [8], DHVR [24], PointDSC [1], PG-Net[40], VBReg[20] . The first 8 methods are based on geometry methods and the last 6 methods are deep learning methods. Note that for deep learning methods, we test them using the provided pre-trained models. Results are shown in Table 1.

**Combined with FPFH.** The putative correspondences generated using FPFH descriptor have an average inlier ratio of 7.34%. As shown in the left column of Table 1, our method outperforms all the other methods in terms of RR, which is the most important criterion. In addition to achieving the best RR, our method also attains lower RE and TE, demonstrating the high accuracy of our alignment. In terms of efficiency, our method also demonstrates its superiority. While RANSAC-100k can achieve an acceptable RR, our method is approximately 58 times faster, while achieving a higher RR and lower errors. Fig. 6 visualizes some scenes, illustrating that our method achieves smaller alignment errors in specific scenarios. Furthermore, our method achieves the highest F1 score, which demonstrates its effectiveness in successfully identifying more inliers for pose estimation.

**Combined with FCGF.** To further validate the generalization performance, we also adopted the learned descriptor FCGF to generate putative correspondences following [1, 5], and reported the registration results. The average inlier ratio at initial set is 24.38%,

**Table 1: Quantitative comparison results on 3DMatch dataset.**

| Method | FPFH (geometry descriptor) | | | | | | | FCGF (learned descriptor) | | | | | | |
|---|---|---|---|---|---|---|---|---|---|---|---|---|---|---|
| | IP (%↑) | IR (%↑) | **F1 (%↑)** | RE (°↓) | TE (cm↓) | **RR (%↑)** | Time (s) | IP (%↑) | IR (%↑) | **F1 (%↑)** | RE (°↓) | TE (cm↓) | **RR (%↑)** | Time (s) |
| FGR[55] | - | - | - | 4.96 | 10.25 | 40.91 | 0.40 | - | - | - | 2.90 | 8.41 | 78.93 | 0.89 |
| SM[25] | 47.96 | 70.69 | 50.70 | 2.94 | 8.15 | 55.88 | 0.03 | 81.44 | 38.36 | 48.21 | 2.29 | 7.07 | 86.57 | 0.03 |
| TEASER[44] | 73.01 | 62.63 | 66.93 | 2.48 | 7.31 | 75.48 | 0.11 | 82.43 | 68.08 | 73.96 | 2.73 | 8.66 | 85.77 | 0.11 |
| GC-RANSAC[2] | 48.55 | 69.38 | 56.78 | 2.33 | 6.87 | 67.65 | 0.62 | 64.46 | 93.39 | 75.69 | 2.33 | 7.11 | 92.05 | 0.47 |
| RANSAC-1k[14] | 51.52 | 34.31 | 39.23 | 5.16 | 13.65 | 40.05 | 0.08 | 76.86 | 77.45 | 76.62 | 3.16 | 9.67 | 86.57 | 0.08 |
| RANSAC-10k[14] | 62.43 | 54.12 | 57.07 | 4.35 | 11.79 | 60.63 | 0.55 | 78.54 | 83.72 | 80.76 | 2.69 | 8.25 | 90.70 | 0.58 |
| RANSAC-100k[14] | 68.18 | 67.40 | 67.47 | 3.55 | 10.04 | 73.57 | 5.24 | 78.38 | 85.30 | 81.43 | 2.49 | 7.54 | 91.50 | 5.50 |
| GROR [43] | 72.54 | 76.08 | 74.10 | 2.22 | 6.89 | 80.78 | 0.46 | 80.01 | 86.36 | 82.80 | 2.00 | 6.48 | 92.67 | 0.46 |
| SC$^2$-PCR [5] | 71.98 | 77.86 | 74.80 | 2.15 | 6.69 | 83.67 | 0.10 | 79.94 | 87.15 | 83.09 | 2.10 | 6.48 | 93.04 | 0.10 |
| MAC[52] | - | - | - | 1.89 | 6.20 | 83.65 | 0.13 | - | - | - | 1.86 | 6.15 | 93.28 | 0.13 |
| 3DRegNet[29] | 28.21 | 8.90 | 11.63 | 3.75 | 9.60 | 26.31 | 0.05 | 67.34 | 56.28 | 58.33 | 2.74 | 8.13 | 77.76 | 0.05 |
| DGR[8] | 28.80 | 12.42 | 17.35 | 3.78 | 10.80 | 69.13 | 2.49 | 67.47 | 78.94 | 72.76 | 2.40 | 7.48 | 91.30 | 1.36 |
| DHVR[24] | 60.19 | 64.90 | 62.11 | 2.78 | 7.84 | 67.10 | 0.46 | 80.20 | 78.15 | 78.98 | 2.25 | 7.08 | 91.93 | 0.46 |
| PointDSC[1] | 68.63 | 71.63 | 69.89 | 2.09 | 6.59 | 78.56 | 0.09 | 79.07 | 86.48 | 82.31 | 2.05 | 6.54 | 93.22 | 0.09 |
| PG-Net[40] | 72.48 | 77.19 | 74.59 | 2.08 | 6.56 | 82.62 | 0.17 | 79.48 | 86.88 | 82.72 | 2.05 | 6.53 | 93.28 | 0.17 |
| VBReg[20] | 72.04 | 74.44 | 73.01 | 2.14 | 6.77 | 82.75 | 0.22 | 79.68 | 87.12 | 83.21 | 2.05 | 6.53 | 93.35 | 0.22 |
| Ours | 75.64 | 82.27 | 78.05 | 2.05 | 6.50 | 86.49 | 0.09 | 88.88 | 95.59 | 90.32 | 2.01 | 6.32 | 94.10 | 0.09 |

**Table 2: Quantitative comparison results on 3DLoMatch dataset.**

| Method | FCGF(learned descriptor) | | | | | | |
|---|---|---|---|---|---|---|---|
| | IP↑ | IR↑ | **F1↑** | RE↓ | TE↓ | **RR↑** | Time(s) |
| SC$^2$-PCR[5] | 44.87 | 53.69 | 48.38 | 3.77 | 10.46 | 57.83 | 0.11 |
| MAC[52] | - | - | - | 3.50 | 9.75 | 59.85 | 0.15 |
| DGR[8] | 42.22 | 38.96 | 39.05 | 4.17 | 10.82 | 43.80 | 1.48 |
| DHVR[24] | 41.96 | 38.60 | 39.22 | 4.14 | 12.56 | 54.41 | 3.55 |
| PointDSC[1] | 44.51 | 52.38 | 47.57 | 3.87 | 10.39 | 56.09 | 0.10 |
| PG-Net[40] | 45.65 | 53.38 | 48.12 | 3.82 | 10.42 | 57.12 | 0.19 |
| VBReg[20] | 45.81 | 54.08 | 49.09 | 3.81 | 10.69 | 58.00 | 0.52 |
| Ours | 50.30 | 66.65 | 54.54 | 3.67 | 10.17 | 61.85 | 0.09 |

| Method | Predator(learned descriptor) | | | | | | |
|---|---|---|---|---|---|---|---|
| | IP↑ | IR↑ | **F1↑** | RE↓ | TE↓ | **RR↑** | Time(s) |
| SC$^2$-PCR[5] | 56.98 | 67.47 | 61.08 | 3.46 | 9.58 | 69.46 | 0.11 |
| MAC[52] | - | - | - | 3.35 | 9.50 | 69.92 | 0.15 |
| DGR[8] | 51.38 | 54.24 | 51.62 | 3.19 | 10.01 | 59.46 | 1.48 |
| DHVR[24] | 54.75 | 54.66 | 53.70 | 4.97 | 12.33 | 65.41 | 3.55 |
| PointDSC[1] | 56.55 | 67.52 | 60.82 | 3.43 | 9.60 | 68.89 | 0.10 |
| PG-Net[40] | 56.67 | 67.86 | 60.92 | 3.52 | 9.73 | 69.02 | 0.19 |
| VBReg[20] | 58.63 | 68.58 | 62.53 | 3.37 | 9.52 | 69.95 | 0.33 |
| Ours | 66.40 | 79.77 | 68.88 | 3.30 | 9.42 | 71.12 | 0.07 |

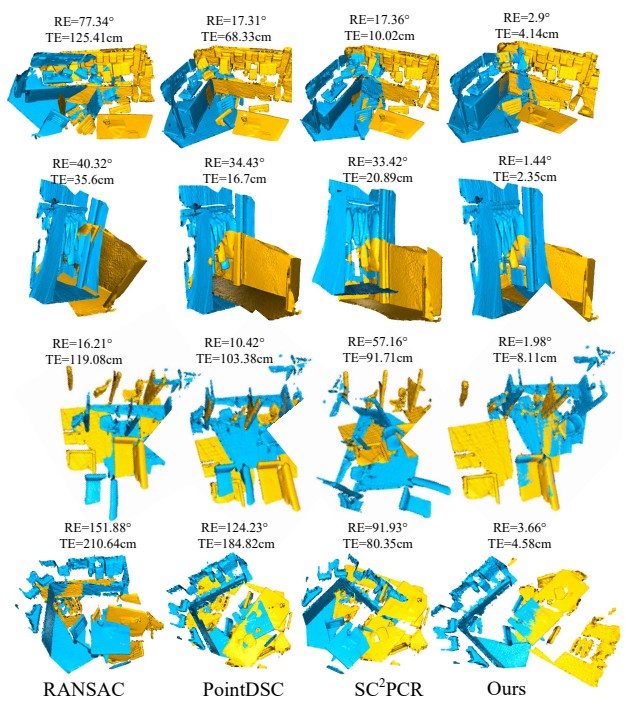

**Figure 6: Registration results on 3DMatch and 3DLoMatch datasets. Suggested color zoom.**

higher than the initial matches obtained by FPFH descriptors. As shown in the right column of Table 1. The performance of all methods has been improved. Our RR remains high at 94.10%, still surpassing all other methods. In addition, our method again demonstrates an acceptable efficiency in terms of the average registration time for a pair of point clouds. It is noteworthy that our progressive correspondence pruning network achieves a 7.11% higher F1 score than VBReg, which convincingly demonstrates the ability to accurately define inliers under different initial inlier ratios.

**Under low-overlapping.** Furthermore, we report the results on sparsely overlapping scenes using 3DLoMatch. Following [1, 5], we utilize FCGF and Predator [19] descriptors to generate correspondences. As shown in Table 2, both FCGF and Predator descriptors combined with our method achieve the highest RR and F1 scores. Moreover, our method achieves the optimal efficiency. This indicates the robustness of our method in low-overlap scenes (10%-30%

overlap) as well as its generalization on novel baseline matchers. Fig. 6 visualizes some scenes, illustrating that our method achieves better alignment.

## 4.3 Results on Outdoor Scenes

The results comparison of RANSAC [14] (100k iterations), SC$^2$-PCR [5], MAC [52], DGR [8], PointDSC [1], PG-Net[40], VBReg[20] on the KITTI dataset is reported in Table 3. Our method achieves higher RR compared to both geometry and learning-based methods

**Table 3: Quantitative comparison results on the KITTI dataset.**

| Method | FPFH (geometry descriptor) | | | | | | |
|---|---|---|---|---|---|---|---|
| | IP↑ | IR↑ | **F1↑** | RE↓ | TE↓ | **RR↑** | Time(s) |
| RANSAC-100k[14] | 78.50 | 70.66 | 74.37 | 1.22 | 25.88 | 89.37 | 13.7 |
| SC²-PCR [5] | 93.63 | 95.89 | 94.63 | 0.32 | 7.23 | 99.64 | 0.31 |
| MAC[52] | - | - | - | 0.40 | 8.46 | 99.46 | 0.23 |
| DGR[8] | 78.39 | 54.12 | 62.15 | 1.64 | 33.10 | 77.12 | 2.29 |
| PointDSC[1] | 89.72 | 86.33 | 87.79 | 0.35 | 7.16 | 98.20 | 0.39 |
| PG-Net[40] | 92.32 | 94.65 | 93.61 | 0.32 | 7.02 | 99.32 | 0.75 |
| VBReg[20] | 91.43 | 94.10 | 93.26 | 0.32 | 7.17 | 98.92 | 0.51 |
| Ours | 93.12 | 96.65 | 94.33 | 0.30 | 7.14 | 99.48 | 0.28 |
| Method | FCGF (learned descriptor) | | | | | | |
| | IP↑ | IR↑ | **F1↑** | RE↓ | TE↓ | **RR↑** | Time(s) |
| RANSAC-100k[14] | 83.62 | 85.77 | 84.68 | 0.38 | 22.60 | 98.38 | 13.4 |
| SC²-PCR [5] | 82.01 | 91.03 | 85.90 | 0.33 | 20.95 | 98.20 | 0.31 |
| MAC[52] | - | - | - | 0.34 | 19.34 | 97.84 | 0.23 |
| DGR[8] | 72.19 | 78.06 | 75.13 | 0.34 | 21.70 | 98.20 | 2.29 |
| PointDSC[1] | 81.99 | 90.69 | 85.74 | 0.33 | 20.90 | 97.66 | 0.40 |
| PG-Net[40] | 81.99 | 90.69 | 85.96 | 0.32 | 20.90 | 98.28 | 0.74 |
| VBReg[20] | 81.33 | 90.21 | 85.56 | 0.35 | 20.91 | 98.02 | 0.54 |
| Ours | 90.48 | 97.81 | 93.32 | 0.35 | 20.59 | 98.52 | 0.28 |

when combined with FCGF. It is noteworthy that SC²-PCR and MAC, which exhibit superior performance on the FPFH descriptor, experience a decline in performance when combined with FCGF. This finding demonstrates the superior generalization capability of our method across different descriptors. Our method shows more noticeable performance in outlier removal. It is worth noting that outdoor point clouds are inherently sparse, and the effectiveness of different combinations of methods and descriptors tends to saturate. Fig. 7 visualizes some scenes.

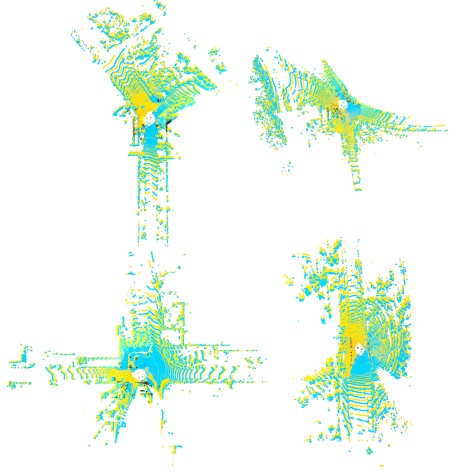

**Figure 7: Registration results on KITTI dataset. Suggested color zoom.**

### 4.4 Multi-way Registration

To further evaluate the performance of our method in multi-way registration, we utilize the Augmented ICL-NUIM dataset. It is worth noting that we test the model trained on the 3DMatch dataset

**Table 4: ATE(cm, Lower is better.) on Augmented ICL-NUIM**

| Method | Living1 | Living2 | Office1 | Office1 | AVG |
|---|---|---|---|---|---|
| ElasticFusion[41] | 66.61 | 24.33 | 13.04 | 35.02 | 34.75 |
| InfiniTAM[21] | 46.07 | 73.64 | 113.8 | 105.2 | 85.68 |
| BAD-SLAM[35] | fail | 40.41 | 18.53 | 26.34 | - |
| Multiway + FGR[55] | 78.97 | 24.91 | 14.96 | 21.05 | 34.98 |
| Multiway + RANSAC[14] | 110.9 | 19.33 | 14.42 | 17.31 | 40.49 |
| Multiway + SC²-PCR[5] | 18.68 | 14.31 | 14.63 | 11.95 | 14.90 |
| Multiway + DGR[8] | 21.06 | 21.88 | 15.76 | 11.56 | 17.57 |
| Multiway + PointDSC[1] | 20.25 | 15.58 | 13.56 | 11.30 | 15.18 |
| Multiway + Ours | 17.93 | 15.10 | 13.20 | 10.06 | 14.07 |

to validate cross-dataset generalization. Following [8] , multi-way registration first extracts FPFH descriptors for each frame, and then employs our proposed method to obtain initial poses through pairwise registration. After that, the poses is globally optimized using the pose graph optimization [22] implemented in Open3D [56]. We report the results of SC²-PCR and the baseline methods proposed in [8], [1]. Absolute trajectory error (ATE) and average results are reported for each scene. As shown in Table 4, our method achieves the lowest average ATE among the four tested scenes.

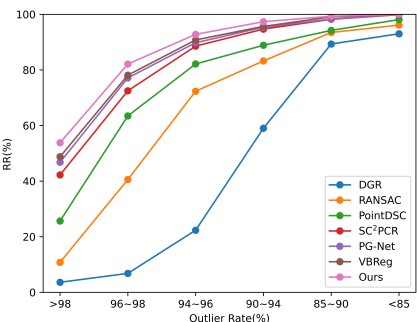

**Figure 8: The RR under the different outlier ratio of the putative correspondences.**

### 4.5 Robustness Test

To verify the robustness of our method at high outlier rate, we report the RR at different outlier rate. Specifically, we generate initial correspondences using FPFH descriptors in the 3DMatch dataset. Based on the outlier rate of each pair of point clouds, the test set is categorized into six groups: <85%, 85%-90%, 90%-94%, 94%-96%, 96%-98%, and >98%. The number of point cloud pairs in each group is 156, 193, 319, 262, 343 and 350, respectively. As shown in Fig. 8, when the outlier rate is >98%, our method outperforms other methods, especially DGR and PointSDC which are also learning-based methods.

### 4.6 Analysis Experiments

In this section, we perform ablation studies and analyze experiments on 3DMatch and KITTI datasets. The PointDSC using an SCNonlocal module is utilized as our baseline, shown in the Row 1 and Row 8 of Table 5. We progressively add the method proposed in

**Table 5: Analysis experiments on 3DMatch / KITTI. Baseline: PointDSC utilizes a SCNonlocal [1] module. DGC: Deep Geometric Coherence. HP: Hypothesis Proposer. 2/3/4-th: The number of pruning blocks.**

| | | Baseline | DGC | HP(knn) | HP | 2-th | 3-th | 4-th | RR (%↑) FPFH | FCGF | Time(s) |
|---|---|---|---|---|---|---|---|---|---|---|---|
| 3DMatch | 1) | ✓ | | | | | | | 77.63 | 92.73 | 0.05 |
| | 2) | ✓ | ✓ | | | | | | 81.63 | 93.41 | 0.05 |
| | 3) | ✓ | ✓ | ✓ | | | | | 81.89 | 93.41 | 0.05 |
| | 4) | ✓ | ✓ | | ✓ | | | | 83.88 | 93.59 | 0.05 |
| | 5) | ✓ | ✓ | | ✓ | ✓ | | | 85.23 | 93.74 | 0.07 |
| | 6) | ✓ | ✓ | | ✓ | ✓ | ✓ | | 86.49 | 94.10 | 0.09 |
| | 7) | ✓ | ✓ | | ✓ | ✓ | ✓ | ✓ | 84.67 | 93.56 | 0.11 |
| KITTI | 8) | ✓ | | | | | | | 98.02 | 97.22 | 0.22 |
| | 9) | ✓ | ✓ | | | | | | 98.68 | 97.76 | 0.22 |
| | 10) | ✓ | ✓ | ✓ | | | | | 98.70 | 97.78 | 0.22 |
| | 11) | ✓ | ✓ | | ✓ | | | | 98.89 | 97.98 | 0.22 |
| | 12) | ✓ | ✓ | | ✓ | ✓ | | | 99.20 | 98.20 | 0.25 |
| | 13) | ✓ | ✓ | | ✓ | ✓ | ✓ | | 99.48 | 98.52 | 0.28 |
| | 14) | ✓ | ✓ | | ✓ | ✓ | ✓ | ✓ | 99.20 | 98.10 | 0.31 |

Sec. 3 to the baseline and report the results in Table 5. The sampling ratio for each pruning is illustrated in Table 6. And we insert our proposed method into the existing learning and geometry methods as shown in Table 7.

**Ablation of backbone.** Firstly, we incorporate DGC learning measure into baseline. By accurately checking the compatibility of correspondences in the measure space, ours method can accurately identify and retain inliers, leading to correct alignment. As shown in Row 1, 2 of Table 5, when incorporating DGC to embed deep consistency, the RR improves by 4% compared to the baseline with FPFH. The Row 8, 9 of Table 5 demonstrate the generalization ability of DGC across datasets. It is easy to understand that DGC, which combines geometric consistency in 3D domain and captures long-range dependencies, can accurately understand different 3D scenes and reduce ambiguity. Secondly, our Hypothesis Proposer can identify an accurate consensus set for each reliable correspondence relationship, leading to accurate hypothesis verification results. As shown in Row 2, 4 and Row 9, 11 of Table 5, the insertion of HP further enhances the RR. It is worth noting that utilizing k-nearest neighbor searching in the DGC feature space for each reliable correspondence yields inferior performance compared to second order spatial compatibility (SC$^2$) searching, as illustrated in Row 3, 4 and Row 10, 11 of Table 5. SC$^2$ searching effectively employs distance and angle constraints to precisely identify consistent consensus sets for each reliable correspondence relationship, leading to more robust transformation estimation results.

**Progressive Pruning.** We analyze the effect of the number of prunings on pose estimation. Each sequential pruning operation is represented by two pruning blocks. As illustrated in Fig. 2, our method increased the initial inlier ratio from 3.4% to 98.6% with increasing sequential pruning iterations, demonstrating that outliers in the initial correspondences were effectively filtered out with continuous sequential pruning. As shown in Row 5, 6, 7 and Row 12, 13, 14 of Table 5, RR decreases after using 4-th pruning blocks, corresponding to three sequential pruning. This decline is attributed to excessive pruning, which may also filter out some inliers, hindering robust pose estimation. The imbalance between

inliers and outliers in the initial correspondences is alleviated by progressively removing outliers, facilitating inlier identification.

**Table 6: RR at different sampling ratios for each pruning.**

| Ratio | 3DMatch | | | KITTI | | |
|---|---|---|---|---|---|---|
| | FPFH | FCGF | Time(s) | FPFH | FCGF | Time(s) |
| 100% | 83.53 | 93.35 | 0.16 | 98.89 | 97.98 | 0.42 |
| 75% | 84.76 | 93.41 | 0.13 | 98.90 | 98.20 | 0.37 |
| 50% | 85.56 | 93.59 | 0.11 | 99.10 | 98.10 | 0.32 |
| 25% | 86.49 | 94.10 | 0.09 | 99.48 | 98.52 | 0.28 |
| 10% | 85.23 | 93.28 | 0.08 | 99.10 | 97.98 | 0.25 |

**Pruning ratio.** As shown in Table 5, we sequentially perform four coherence pruning blocks, using specific sampling ratio to progressively prune the initial correspondences $C$ into a candidate subset $C_m$. The results of different sampling ratios are presented in Table 6, where the performance and efficiency are deemed acceptable when the sampling ratio is 25%. This implies that 75% of the network-identified outliers are filtered out with each pruning.

**Table 7: The RR of combining DGC learning measure and pruning strategy with learned and geometry based methods.**

| Method | 3DMatch | | KITTI | |
|---|---|---|---|---|
| | FPFH | FCGF | FPFH | FCGF |
| PointDSC[1] | 78.56 | 93.22 | 98.20 | 97.66 |
| + DGC | 83.32 | 93.47 | 98.76 | 98.20 |
| + 3-th | 82.86 | 93.35 | 98.57 | 98.10 |
| + DGC + 3-th | 84.56 | 93.71 | 99.21 | 98.32 |
| SC$^2$-PCR[5] | 83.67 | 93.04 | 99.64 | 98.20 |
| + DGC | 85.27 | 93.74 | 99.64 | 98.32 |
| + 3-th | 85.12 | 93.59 | 99.64 | 98.30 |
| + DGC + 3-th | 86.76 | 94.15 | 99.66 | 98.54 |

**Combined with learning and geometry methods.** We evaluate the flexibility of our proposed method by combining it with deep learning method PointDSC and geometry method SC$^2$-PCR. The results are shown in Table 7. Ours method significantly improves the RR for all tested methods on the 3DMatch and KITTI datasets. It is worth noting that SC$^2$-PCR achieves state-of-the-art RR performance when enhanced by ours method.

## 5 CONCLUSION

To mitigate the detrimental impact of numerous outliers, we propose a lightweight progressive 3D correspondence pruning network for accurate and efficient point cloud registration. Our method achieves excellent performance on public benchmarks. Furthermore, it can be combined with both learned and geometry registration methods to boost their performance.

**Limitation.** A higher initial inlier ratio is associated with superior registration performance. Future work will develop more robust feature descriptors to enhance inlier ratio. In addition, the current framework employs a constant sampling rate. Future work will investigate a adaptive-ratio pruning strategy.

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
