# OpenReview forum: "3DPCP-Net: A Lightweight Progressive 3D Correspondence Pruning Network for Accurate and Efficient Point Cloud Registration"
_acmmm.org/ACMMM/2024/Conference — MM2024 Poster_

### Official Review · Reviewer_W4Xq · 2024-05-01

**Rating:** 1
**Confidence:** 4

**Summary:**

This paper proposes a lightweight network for fast and robust registration. This paper introduces a DGC learning block to examine the compatibility of two correspondences by exploring deep feature similarity and pairwise spatial distances and angles, thereby facilitating the correspondence pruning process.

**Strengths:**

1)	This method is lighter in weight and shorter in time.
2)	This method is highly versatile and can be easily integrated into both learning-based and geometry-based frameworks.

**Limitations:**

1)	The comparison algorithm selected on the 3DMatch/3DLoMatch dataset does not include the current SOTA algorithm, and has not been compared with the current optimal algorithm GeoTransformer/RegTR and other algorithms.
2)	The paper says it is easy to integrate with other models. However, the number of other models combined in the experimental setting is too small, and it is not combined with the current optimal algorithm.

**Suitability:**

2

---

### Official Review · Reviewer_3u8y · 2024-05-09

**Rating:** 5
**Confidence:** 2

**Summary:**

This paper focuses on inlier/outlier classification of initial 3D correspondences for point cloud registration. This paper presents a lightweight progressive framework to efficiently prune the correspondences. This paper also proposes deep geometric coherence features to further enhance the performance. The various experiments on 3DMatch, 3DLoMatch, KITTI and Augmented ICL-NUIM datasets demonstrate the efficiency of the method, w.r.t. both accuracy and speed.

**Strengths:**

1.	This manuscript provides comprehensive evaluations with different settings (indoor/outdoor, different descriptors, etc) to demonstrate the efficiency of the proposed method.
2.	As a progressive method, the proposed 3DPCP-Net does not sacrifice speed for accuracy. Instead, it runs much faster and requires much fewer network parameters, as shown in Figure 1 and experiment result tables.
3.	The proposed method can be easily integrated into existing methods to enhance the performance, as demonstrated in Table 7.
4.	Various visualizations are presented to support the efficiency (both accuracy and speed) of the proposed method. The manuscript is well-written and easy to read.

**Limitations:**

1.	About the main contribution: a lightweight progressive pruning method to improve efficiency. More details could be provided to support its advantages.

   (1)	The fast speed is mentioned in the introduction and experiments. In the method part (Sec.3), however, it is unclear why the progressive method could be much faster than one-shot methods. This non-trivial contribution should be discussed in detail.

   (2)	Reference [40] (mentioned in Line 249) also adopts a progressive design. What is the difference between [40] and the 3DPCP-Net?

2.	About the number of pruning blocks.

   (1)	This detail could be explicitly listed in Implementation Details of Section 4.1.

   (2)	Table 6: are the results generated with 3 pruning blocks or 4?

3.	Figure 4 does not correspond to Lines 398-439 well. For example, where is “gc”, and why are there two “a” (or alpha) but only one “d”?

4.	Figure 6 contains visualization of two datasets. For each row, the details could be listed, including the dataset, the descriptor for initial correspondence generation, etc.

**Suitability:**

2

---

### Official Review · Reviewer_PANt · 2024-05-24

**Rating:** 3
**Confidence:** 2

**Summary:**

This paper proposes the 3DPCP-Net for fast and robust registration. Its core design lies in progressive correspondence pruning through mining deep spatial geometric coherence, which can effectively learn pairwise 3D spatial distance and angular features to progressively remove outlier (mismatched correspondence) for accurate pose estimation. 3DPCP-Net also contains an efficient feature-based hypothesis proposer that leverages the geometric consistency features to generate reliable model hypotheses for each reliable correspondence explicitly. Extensive experiments demonstrate the accurate and efficient for outlier removal and pose estimation tasks.

**Strengths:**

+ 3DPCP-Net progressively prunes the set of correspondences, rather than a one-shot classification of initial correspondences.
+ Since most outliers are filtered out after the progressive pruning, 3DPCP-Net allows to identify reliable inliers among the remaining entities, which leads to accurate pose estimation.
+ Explicitly and progressively pruning the initial correspondences, as opposed to purely using deeper networks, does not result in an increase in network parameters.
+ 3DPCP-Net contains a feature-based hypothesis proposer that leverages the high-dimensional spatial geometric consistency features extracted above and spatial geometric consistency searching to efficiently generate multiple reliable hypotheses.
+ Benefiting from the invariance of spatial distance and angle under rigid transformation between inliers and the feature similarity, DGC can obtain more representative features for each correspondence, facilitating accurate inlier/outlier differentiation.

**Limitations:**

# Method section
+ 3DPCP-Net contains multiple stacked Pruning Blocks. I would like to know the difference between the Pruning Block and (SCNonlocal module + Seed Selection + NMS module + LS + Hypothesis) in PointDSC? It seems that the DGC module in 3DPCP-Net introduces angle constraints in the SCNonlocal module in PointDSC.

**Suitability:**

2

---

### Official Review · Reviewer_xVki · 2024-05-24

**Rating:** 2
**Confidence:** 4

**Summary:**

This paper proposes a  lightweight network, 3DPCP-Net, to filter the outlier s of the point-to-point correspondence between source and target point cloud. And the experiments on variosu dataset demonstrate that it does work on these data. However, this is a data-driven method, and it is possible that the trained network overfits.

**Strengths:**

1) The writing is easy to read.
2) The proposed correspondence filtering method does work on the selected datasets.

**Limitations:**

1) The novelty and the contribution is limited. The whole paper propse a learned correspondence filtering method, where the input is Nx6 tensor, and the output is the confidence score. However, previous methos already proposed similar module, such as Pointdsc. The difference is that the details of the module.
2) On some datasets, the proposed method is worse than MAC.
3) Can the proposed method guarantee the spatial smoothness of the correspondence?

**Suitability:**

2

---

### Meta-Review · Area_Chair_JNvd · 2024-07-03

**Recommendation:** Accept (Poster)
**Confidence:** 3

**Metareview:**

This paper proposes a lightweight network for fast and robust point cloud registration which progressively filters the outliers of the point-to-point correspondence. The proposed method can be easily integrated into existing methods.

There were some concerns about the SOTA comparisons in the original submission. However, the authors introduced new comparisons that show the effectiveness of the method when compared with those in the rebuttal.